# Effect of sugar metabolite methylglyoxal on equine lamellar explants: An *ex vivo* model of laminitis

**Cristina Vercelli**[ID]*[☯], **Massimiliano Tursi**[☯], **Silvia Miretti**[☯], **Gessica Giusto**[☯], **Marco Gandini**[☯], **Giovanni Re**[☯], **Emanuela Valle**[☯]

Department of Veterinary Science, University of Turin, Grugliasco (TO), Italy

☯ These authors contributed equally to this work.
* cristina.vercelli@unito.it

**Data Availability Statement:** All relevant data are within the manuscript and its Supporting Information files.

## Abstract

Laminitis is one of the most devastating diseases in equine medicine, and although several etiopathogenetic mechanisms have been proposed, few clear answers have been identified to date. Several lines of evidence point towards its underlying pathology as being metabolism-related. In the carbonyl stress pathway, sugars are converted to methylglyoxal (MG)—a highly reactive α-oxoaldehyde, mainly derived during glycolysis in eukaryotic cells from the triose phosphates: D-glyceraldehyde-3-phosphate and dihydroxyacetone phosphate. One common hypothesis is that MG could be synthesized during the digestive process in horses, and excessive levels absorbed into peripheral blood could be delivered to the foot and lead to alterations in the hoof lamellar structure. In the present study, employing an *ex vivo* experimental design, different concentrations of MG were applied to hoof explants (HE), which were then incubated and maintained in a specific medium for 24 and 48 h. Macroscopic and histological analyses and a separation force test were performed at 24 and 48 h post-MG application. Gene expression levels of matrix metalloproteinase (MMP)-2 and -14 and tissue inhibitor of metalloproteinase (TIMP)-2 were also measured at each time point for all experimental conditions. High concentrations of MG induced macroscopic and histological changes mimicking laminitis. The separation force test revealed that hoof tissue samples incubated for 24 h in a high concentration of MG, or with lower doses but for a longer period (48 h), demonstrated significant weaknesses, and samples were easily separated. All results support that high levels of MG could induce irreversible damage in HEs, mimicking laminitis in an *ex vivo* model.

## Introduction

Laminitis is one of the most painful and debilitating diseases in equine medicine, with significant implications in terms of horse welfare. It is characterized by deterioration of the lamellar tissue that connects the hoof wall to the underlying third phalanx in the equine hoof, with associated inflammation, heat, digital pulses, pain and lameness, and, in some cases, dorsopalmar

**Funding:** This study was supported by Ministero dell'Istruzione, dell'Università e della Ricerca (MIUR) under the programme "Dipartimento di Eccellenza ex L.232/2016" to the Department of Veterinary Science, University of Turin (to authors EV, MG, and GG); and by the ex 60% fund of the University of Turin. The funders had no role in study design, data collection and analysis, decision to publish, or preparation of the manuscript.

**Competing interests:** The authors have declared that no competing interests exist.

rotation of the third phalanx [1]. It can be classified as either acute or chronic, and it is not usually readily reversible. Laminitis can lead to permanent disability or even death [1]. The most common etiology of laminitis cases involves gastrointestinal or metabolic disease [2]. For a number of reasons, laminitis is considered a metabolism-related pathology, resembling the human diabetic foot, since hormonal influences, weight and pressure are involved [3–5]. Evidence suggests that diabetic microvascular complications, including neuropathy, are caused by the activation of mechanisms related to the polyol pathway: advanced glycation end-product (AGE) formation with subsequent activation of the receptor for AGEs (RAGE), and lead to the activation of other factors such as nuclear factor kB, protein kinase C (PKC) isoforms, and the hexosamine pathway as a result of the overproduction of superoxide by mitochondria [6].

Gastrointestinal disturbances resulting from ingestion of excess carbohydrates play a pivotal role in sepsis-related laminitis [7, 8]. Feeds rich in starch are used as highly digestible energy sources in horses. During digestion in the small bowel, starch is primarily broken down by amylase enzymes, leading to glucose liberation. Extremely large quantities of starch in the diet can result in starch overload; which means that undigested starch passes from the small to the large intestine, which can lead to a decrease in colon and caecal pH, often followed by colic laminitis [9].

In this scenario, release of toxins from the hindgut is suspected to occur, which in theory induces degradation of the lamellar basement membrane and a loss in epithelial cell adhesion, with the subsequent activation of matrix metalloproteinases (MMPs) and leukocyte infiltration into the lamellae [4]. According to some authors, AGEs accumulate in significant amounts in the hoof lamellar tissue in the acute phase of experimentally induced laminitis using the hyperinsulinemic model [10]. Moreover, Valle et al. [11] revealed increased plasma levels of pentosidine, a glycoxidative marker, in ponies with clinical equine metabolic syndrome. Methylglyoxal (MG) causes the formations of AGEs, leading to carbonyl stress [12]. AGES derived from glucose and intermediates like MG maybe the major source of intracellular and plasma AGEs [13]. Since MG is a highly reactive intermediate, it is converted into D-lactate, to prevent the formation of AGEs.

Specifically, starch overload in the gut can lead to rapid, devastating changes that result in the elaboration of toxins and other substances: Gram-negative and -positive bacteria undergo carbonyl stress, leading to methylglyoxal biosynthesis—a highly reactive α-oxoaldehyde, mainly derived from the triose phosphates D-glyceraldehyde-3-phosphate and dihydroxyacetone phosphate, which would be detoxified into D-lactate under healthy physiological conditions [14]. In equines, feeding large amounts of fermentable carbohydrates with subsequent acidosis increases the plasma levels of D- lactate up to 25 mMol/L [13]. According to this mechanism, D-lactate can be considered a clinical marker of the development of laminitis, because it is more specific than its L-isoform and more stable than methylglyoxal [9, 15].

Matrix metalloproteinases (MMPs) have been long investigated for their role in the cleavage of the extracellular matrix, which precedes and enables remodeling of tissues and the selective degradation of membrane proteins and collagen IV and VII [16, 17].

Previous studies using *ex vivo* hoof explant models have shown that under specific conditions, basal epidermal cells can separate from their basement membrane as occurs *in vivo*, making them an attractive model for the study of laminitis. In these investigations, silymarin or LPS were used to stimulate a tissue reaction analogous to laminitis in hoof explants [18, 19].

The aim of the present study was to investigate the effects of MG on equine lamellar explants in an *ex vivo* model by evaluating macroscopic and histological structures, tissue integrity by means of the separation force test, and by quantifying MMP expression at different exposure time points.

## Materials and methods

### Animals

Nine male draft horses (Breton horses), reared for meat production, with an age ranging between 18 and 24 months, were enrolled in the study. No animals were specifically killed for the purposes of this study. Horses were slaughtered in a commercial abattoir in Volpiano (Turin, Italy), and front limbs were collected with the slaughterhouse owner's consent under the control of a supervisor from the Italian National Health Service.

### Sample collection

A complete physical examination of all animals was performed, with particular attention to symptoms and signs of laminitis (i.e.: lameness, typical stance, increased digital pulse, and increased hoof capsular warmth). Following slaughter, the right fore distal limb was disarticulated at the carpal joint, and the limbs were transferred onto ice within 45 minutes and brought to the dissection room of the Department of Veterinary Sciences of the University of Turin (Italy).

### Hoof explant preparation

Hoof explant (HE) collection and treatment procedures are summarized in Fig 1.

Hooves were carefully cleaned using a hoof-knife and scrubbed with brushes and chlorhexidine solution (4%). All instruments were cleaned with chlorhexidine and sterile saline. Ice was used to keep all instruments and samples cool.

Explants were collected according to the method reported in Mungall and Pollitt [16] with slight modifications. Hooves were trimmed, removing the lateral and medial walls and cutting the digits into 4–5 sagittal slices. In this way, it was possible to obtain hoof wall strips measuring 0.8x1.5 cm in thickness containing 10–12 lamellae, consisting of the inner part of the hoof wall epidermal lamellae, dermal lamellae and the bone. Prior to incubation, HEs were washed three times with Phosphate Buffered Saline (PBS) under sterile conditions.

### *Ex vivo* tissue survival conditions–preliminary assay

Before the execution of the present project, a preliminary assay was performed to identify the best culture medium for preserving HEs: the tests were run to evaluate the use of Dulbecco's

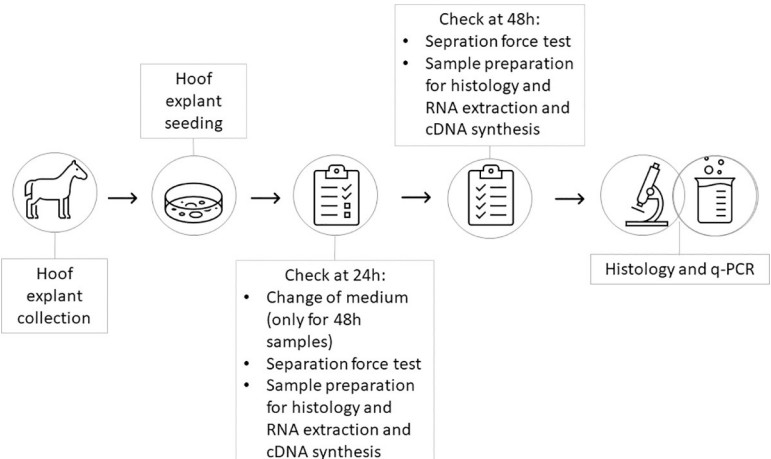

**Fig 1. Timeline.** The image represents the timeline of the experimental phases.

Modified Eagle Medium (DMEM) alone (with 4.5mg/L glucose, but without L-glutamine) *vs* DMEM plus 2% of antibiotic/antimycotic solution, 2% of L-glutamine, and 10% Fetal Bovine Serum (FBS) (DMEM+). All reagents were purchased by Sigma Aldrich (Milan, Italy).

The HEs were incubated in 5 mL of their assigned medium in 6 well-plates. Plates were incubated at 37˚C (5% $CO_2$ and 95% $O_2$ atmosphere) for 24 or 48 hours. The medium in the plates assigned to the 48 hour incubation condition was changed after 24 hours.

### *Ex vivo* tissue survival conditions

Sample survival was evaluated considering macroscopic features (changing of colour, smell, and disruption) with histology as confirmation. In accordance with the results obtained from the preliminary assay (see details in "Results" section), the DMEM+ medium was used for the rest of the study.

### Experimental design

HEs collected from nine different horses were incubated in triplicate with DMEM+ medium in six-well plates. MG was added to the HEs at the following concentrations: 50, 100, 150 and 200 mM, and incubated at 37˚C (5% $CO_2$ and 95% $O_2$ atmosphere) for 24 and 48 hours. HEs cultured in the absence of MG were used as controls (K). In order to allow the execution of the entire experimental design, 45 samples were incubated.

### Separation force test

The separation force test was used to evaluate structural integrity of the samples. All tests were performed in a blinded manner. All HEs were evaluated using a strain gauge (Fig 2) as follows: one end of the HE was fixed to a tong and the other was attached to a force transducer and exposed to a maximum force of 4000 x *g*. Separation force was averaged over nine experiments for each time point (24 and 48 h) by the same operator. Only HEs that separated at the level of epidermal lamellae or dermal lamellae were considered as valid measurements. The HEs that were not broken at these points were considered virtually impossible to separate. The procedure was coordinated in order to immediately proceed with RNA extraction (details in section "RNA extraction and cDNA synthesis").

### Histology

One sample from each time point and from each MG concentration was fixed in 4% buffered formalin. These samples were then embedded in paraffin, and 3-μm slices were mounted on

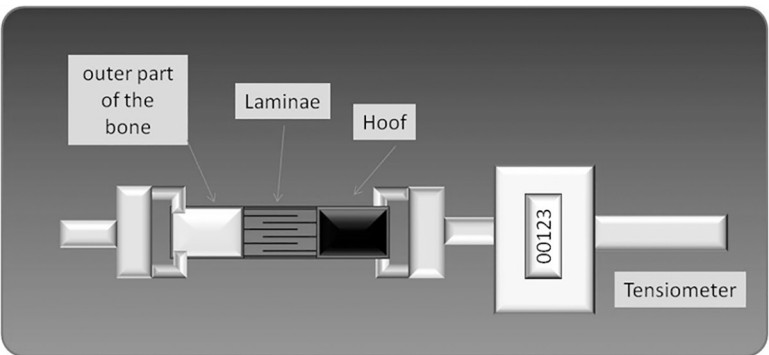

**Fig 2. Separation force test.** The image depicts the equipment used to perform the separation force test.

silanized glass slides. Samples were air-dried and stained histochemically with haematoxylin and eosin (H&E) and periodic acid-Schiff (PAS) (Dako, Milan, Italy).

Slides were examined using a Nikon microscope, and images captured and stored with Image pro-plus software (Media Cybernetics, MD, USA). The same expert operator performed all evaluations, blinded to treatment group. The laminitic lesions were scored using the histopathological grading system proposed by Pollitt [20]. Briefly, lesions of epidermal lamellae attributable to laminitis induced by the experimental design were graded using the following evaluation method: normal (grade N), mild (Grade 1), moderate (Grade 2), severe or extensive (Grade 3). The grading system was based on the degree of change observed in the lamellar basement membrane. Samples were then evaluated for lesions of the secondary epidermal lamellae [20].

### RNA extraction and cDNA synthesis

Immediately after the separation force test, hoof explants (HEs) were placed in tubes containing RNAlater solution (Ambion), and the *stratum lamellatum* was disrupted using a TissueLyser II (Qiagen) with stainless steel beads in 1 mL TRI Reagent (Sigma Aldrich). Total RNA from tissue samples was extracted and purified from any residual genomic DNA using a DNAse I Recombinant RNAse free kit (Roche, Mannheim, Germany). RNA concentration was measured using spectrophotometry (BioPhotometer, Eppendorf, Germany). The ratio of the optical densities measured at 260 and 280 nm was greater than 1.9 in all RNA samples. cDNA was synthesized from 1 μg of total RNA using a RT high Capacity cDNA Reverse Transcription kit (Applied Biosystems, Foster City, CA, USA) according to the manufacturer's protocol. The cDNA was subsequently diluted in nuclease-free water and stored at -20˚C.

### Quantitative assessment of MMP gene expression

Sufficient cDNA was prepared in a single run to perform the real-time quantitative PCR (qPCR) experiments for all the selected genes. To determine the relative amount of specific MMP-2, MMP-14, and TIMP-2 transcripts, qPCR was performed using the CFX Connect Real-time System (Bio-Rad, Hercules, CA, USA). Primers for target and reference genes were designed for *Equus caballus* GenBank mRNA sequences using Primer 3 Software (version 4.0). In order to minimize the amplification of contaminant genomic DNA, oligonucleotides were designed using exon/exon boundaries analyzed using the IDT tool (available at http://www.idtdna.com/scitools/scitools.aspx) for hairpin structure and dimer formation. Primer specificity was verified using BLAST analysis, comparing results against the genomic NCBI database. Table 1 reports primer sequences, gene accession numbers, and amplicon sizes. To establish primer efficiency, we used the dilution method: CFX Manager software (version 3.0, Bio-Rad, Milan, Italy) was used to calculate primer efficiency using the linear regression slope of the dilution series. Each primer set efficiency was between 95% and 100% [21]. Multiple housekeeping genes were selected as potential internal control mRNAs: ß-2 microglobulin (B2M), ß-actin (ACTB), and glyceraldehyde 3-phosphate dehydrogenase (GAPDH). Based on the stable cycles quantification (Cq) values among the different experimental conditions, GAPDH was selected as the most suitable internal control. The q-PCR parameters were as follows: 95˚C for 3 min, 95˚C for 15 sec, and 60˚C for 30 sec, for a total of 44 cycles. To evaluate mRNA expression, data obtained as Cq values of technical replicates were averaged and used to determine ΔCq values (ΔCq = Cq of the target gene–Cq of the reference gene). The ΔΔCq method was used to analyze the data, and results were expressed as fold changes compared with control samples [22]. Assays were run in triplicate, and template control was included using water instead of cDNA.

**Table 1. Genes subjected to quantitative PCR (q-PCR) analysis.**

| Gene name | Sequence 5'-3' | Amplicon size (bp) | Gene bank accession number |
|---|---|---|---|
| MMP-2 | TTCTTCTTCAAGGACCGGTTCA | 93 | XM_001493281 |
| | GAGCTCAGGCCAGAATGTGG | | |
| MMP-14 | CCTGATAAGCCCAAAAACC | 209 | XM_005603265 |
| | CTTCCTCTCATAGGCAGTGTT | | |
| TIMP-2 | TCTACGGCAACCCCATCAAG | 101 | XM_005597879 |
| | GGAGGGAGCCGTGTAGATGA | | |
| GAPDH | TGTCAGCAATGCCTCCT | 191 | NM_001163856 |
| | AAGCAGGGATGATGTT | | |
| β-ACTIN | CATCCGTAAGGACCTGT | 189 | NM_001081838 |
| | GTGGACAATGAGGCCA | | |
| β-2 MICROGLOBULIN | ACCCAGCAGAGAATGGAAAGC | 93 | NM_001082502 |
| | CATCTTCTCTCCATTCTTTAG | | |

Primer sequences for q-PCR were designed for *Equus Caballus* GenBank messenger RNA sequences. Primer sequences, gene accession numbers and amplicon sizes are shown.

## Statistical analysis

Results were organized using Excel software (Microsoft Corporation, CA, USA) and data were analyzed with Prism 9.0 software (Graph Pad, CA, USA).

The results of the separation force test and qPCR were analyzed using one-way Anova and Tukey's multiple comparison test.

Histology score data were recorded, and their distributions were analyzed using two-way Anova, setting time and concentration as variables.

The normality of q-PCR data was checked using D'Agostino and Pearson's omnibus normality test.

Results are presented as means ± standard deviation (SD). Statistical significance was accepted at *p* values $\leq 0.05$.

## Results

### Preliminary assay

Samples maintained in DMEM underwent a change of colour (from a white-light pink colour to green) and smell, indicating that autolytic processes were underway. Histological examination was performed for all samples in order to confirm the macroscopic evaluation. The separation force test was only performed in a few samples because the majority of samples already showed clear visual evidence of disrupted lamellae. On the contrary, samples maintained in DMEM+ medium showed no macroscopical changes, and histological evaluation confirmed the integrity of all structures; all samples were considered eligible for the separation force test. These details are shown in Fig 3.

### Macroscopic evaluation

Considering the results obtained in the preliminary assay, samples were kept in DMEM+ for all subsequent experiments. During the incubation period, all the samples were checked daily. No alterations in colour or other macroscopic changes were detected.

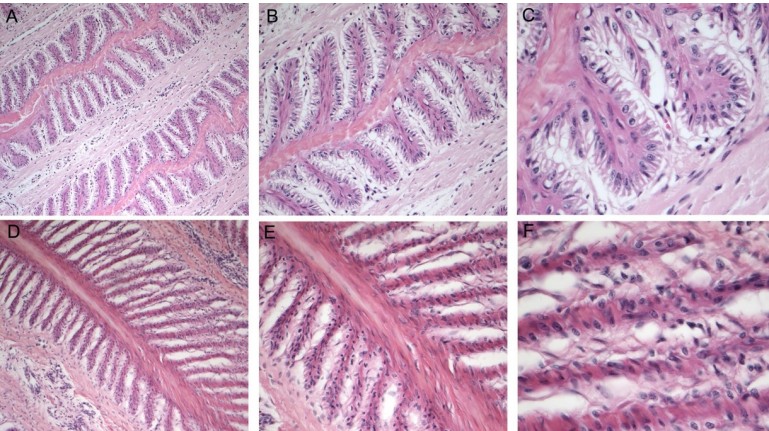

**Fig 3.** A-C, histology of the *stratum lamellatum*, DMEM+. Primary epidermal lamellae and primary dermal lamellae have well-recognizable structure and cellular characteristics. Hematoxylin and eosin (H&E) stain. A: 10 X. B: 20 X. C: 40X. D-F, histology of the *stratum lamellatum*, DMEM. Primary epidermal lamellae and primary dermal lamellae have identifiable but less accurate morphological characteristics than DMEM+ cases. The morphological characteristics of cells and their relationship to contiguous structures are completely compromised (F). H&E stain. D: 10 X. E: 20 X. F: 40 X.

## Separation force test

The separation force test was performed for all samples incubated with the different concentrations of MG at predetermined time points.

The data reported in Table 2 show that after 24 h of incubation in the presence of varying concentrations of MG, separation of the HE lamellar structures could be provoked in 55% of samples (25/45). After 48 h, the percentage increased to 77% (35/45). The representation in Fig 4 shows the averaged weights (kg ± SD) required to provoke separation. No statistically significant effect was observed according to the different MG concentrations, probably due to the high variability of samples. Statistically significant difference can be observed comparing the two time points (24h and 48h). Moreover, a trend for reduced tissue integrity in samples treated with 50, 100 and 150 mM MG for 48 h can be observed.

## Histology

Histological analysis permitted the ranking of samples according to a scoring system, as described above in the Materials and Methods section; the distributions of scores per

**Table 2. Results of separation force test.**

| | Number of explants | | |
|---|---|---|---|
| | Separated/Total | Separated/Total | Separated/Total |
| MG concentration (nM) | T0 | T24 h | T48 h |
| K | 0/9 | 5/9 | 5/9 |
| 50 | - | 4/9 | 8/9 |
| 100 | - | 4/9 | 8/9 |
| 150 | - | 4/9 | 8/9 |
| 200 | - | 8/9 | 6/9 |
| Total | 0/9 | 25/45 | 35/45 |

The table summarizes the results of the separation force test performed on hoof explants (HEs) following 0, 24 and 48 h incubation in different concentrations of MG (total number of samples = 45).

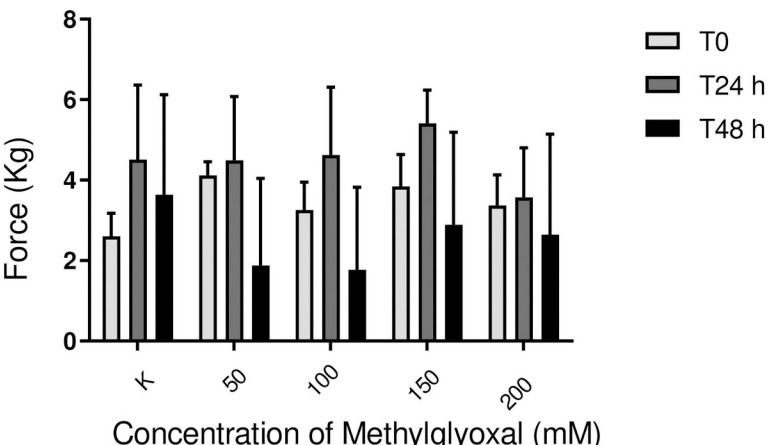

| Tukey's Multiple Comparison Test | Mean Diff. | q | Significant? P < 0.05? | Summary | 95% CI of diff |
|---|---|---|---|---|---|
| T0 vs T24h | -1.084 | 3.428 | No | ns | -2.362 to 0.1937 |
| T0 vs T48 h | 0.8731 | 2.761 | No | ns | -0.4048 to 2.151 |
| T24h vs T48 h | 1.957 | 6.189 | Yes | ** | 0.6793 to 3.235 |

**Fig 4. Distribution of the separation force data.** The image represents the distribution of values obtained in the separation force test prior to (T0) and after (T24 h and T48 h) incubation in different concentrations of methylglyoxal (0, 50, 100, 150 and 200 mM) (N = 45 per time point).

experimental condition are shown in Fig 5. Hoof sections incubated for 48 h in the presence of high concentrations of MG scored the highest due to the alteration of all layers and were statistically different from samples incubated for 24h. In these samples (classified as laminitic stage

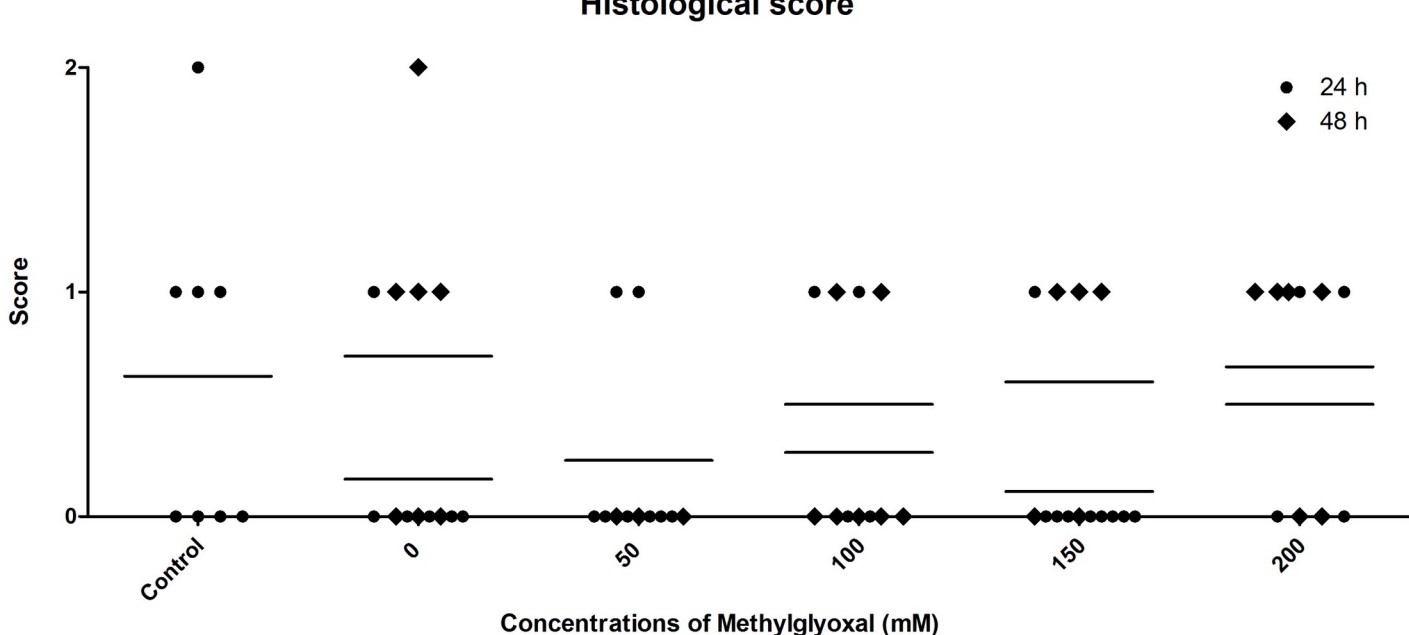

**Fig 5. Distribution of the histological score data.** The graph represents the distribution of histological scores after the incubation of HE in different concentrations of MG (ranging from 0 to 200 mM) for 24 and 48 hours.

2), the following histological changes were observed under low magnification: wavy primary epidermal lamellae; the absence of connective tissue between secondary epidermal lamellae; and rounded basal nuclei. Under high magnification, the round shape of the nuclei was confirmed; basal cells showed a pale, cloudy and vacuolated cytoplasm; and the normal organization of palmar anatomy was no longer recognizable: the definition near the tip of the secondary epidermal lamellae was lost.

## Expression of MMP-2, MMP-14 and TIMP-2 mRNA in digital lamellar specimens

The expression of genes encoding MMPs associated with inflammation, such as MMP-2 and MMP-14, was not substantially increased in digital lamellar samples following 24 h incubation with MG compared with control samples. Concerning MMP-2, higher levels of expression were observed in samples incubated for 48 h compared with controls, with differences also evident for the different MG concentrations. Nevertheless, these differences did not reach the level of statistical significance. This suggests that MG may have a concentration-dependent effect following 48 h incubation. That said, no significant differences between samples, or between the two experimental time points, were detected. Significant differences in TIMP2 expression levels were detected between the MG-treated samples and controls at 24 h at any concentration. At the 48 h time point, the expression levels of TIMP2 were inversely related to MMP-2 and MMP-14 levels. Results are presented in Figs 6–8.

## Discussion

Regarding the primary aim of this study, our results support the use of *ex vivo* equine lamellar hoof wall and dermis explants cultured in DMEM with the appropriate addition of FBS, antibiotic and antimycotic solution, and L-glutamine as a suitable methodology for obtaining a suitable viable model system for investigating the pathological mechanisms of laminitis.

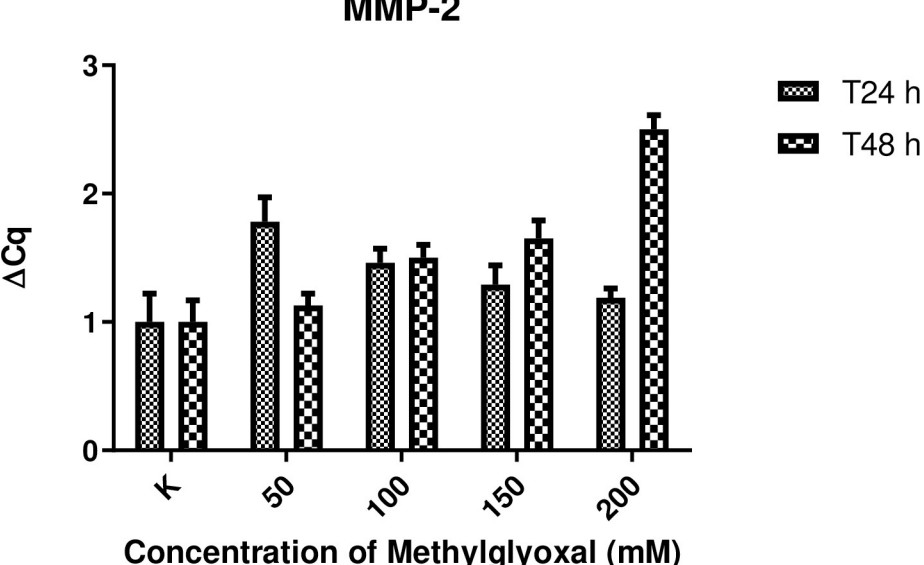

**Fig 6. Effect of MG treatment upon MMP-2 gene expression.** MMP-2 gene expression levels following 24 and 48 h incubation in the presence of increasing concentrations of MG (50, 100, 150 and 200 mM). Control samples (K) were incubated in the absence of MG. Total N = 45 for each time point (24 and 48 h).

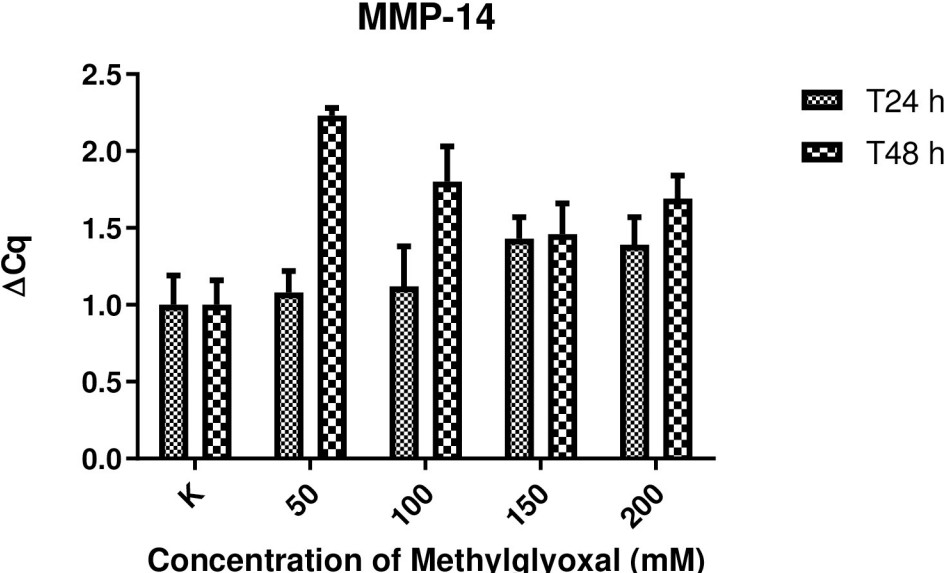

**Fig 7. Effect of MG treatment upon MMP-14 gene expression.** MMP-14 gene expression levels following 24 and 48 h incubation in the presence of increasing concentrations of MG (50, 100, 150 and 200 mM). Control samples (K) were incubated in the absence of MG. Total N = 45 for each time point (24 and 48 h).

Here, we focused on the possible role of MG in laminitis. To this end, macroscopic and histological evaluations, the separation force test, and MMP-2, MMP-14 and TIMP-2 gene expression level assessments were performed.

The preliminary assay was a milestone for the entire experiment: glucose supply was shown to be essential for the maintenance of an integral hoof structure in the explant model, corroborating the results of previous studies that evaluated the integrity of explants in the presence or

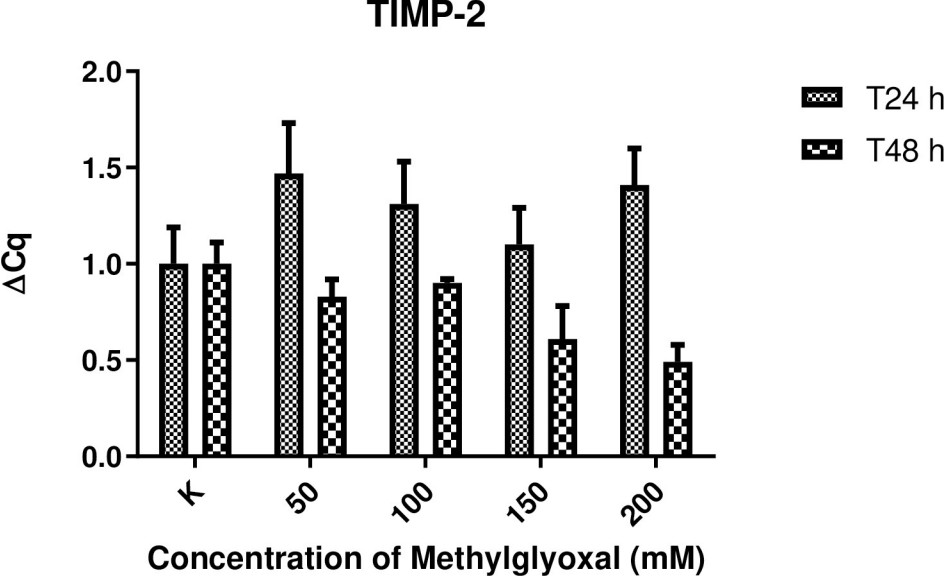

**Fig 8. Effect of MG treatment upon TIMP-2 gene expression.** TIMP-2 gene expression levels following 24 and 48 h incubation in the presence of increasing concentrations of MG (50, 100, 150 and 200 mM). Control samples (K) were incubated in the absence of MG. Total N = 45 for each time point (24 and 48 h).

absence of glucose [23, 24]. In accordance with the method published by Reisinger et al. [25], all explants were cultivated in medium containing high glucose concentrations (4.5 g/L) and antibiotics, but in accordance with our results, FBS and L-glutamine are necessary for tissue preservation. The macroscopic evaluation of explants cultivated in DMEM+ showed the absence of any colour or smell changes throughout the entire experimental period, supporting the viability of the samples for further investigations into the pathogenesis of laminitis. Other papers have evaluated the possibility of using hoof explant samples but focus their attention on different metabolic goals (eg.: the role of insulin) or with the final aim to obtain cell culture. To the authors knowledge this is the first paper focusing specifically on the role of MG [25, 26].

The results obtained in the separation force test support the hypothesis that MG can lead to modification of the hoof lamellae, rendering them less robust and unable to resist separation following the application of a load that untreated samples are able to bear and remain intact. This evaluation confirmed that higher concentrations of MG and a longer incubation period (48 h) can cause irreversible damage that weakens hoof tissues. These results were confirmed by histological examination of HEs, which revealed major alterations in the basal lamella, the severity of which correlated with increasing concentrations of MG.

The results of gene expression of MMPs and TIMP-2 in HEs are in accordance with those obtained by other groups [23, 27]. Increased levels of MMPs seem to be correlated to increasing degrees of extracellular matrix remodeling, more severe histological features, and resulting in structural weakness as assessed by the separation force test. On the contrary, it was expected that TIMP2 levels would decrease in a proportional way as previously shown [16, 28, 29]. MMPs are important for maintaining the integrity of healthy hoof tissue [16, 28]. Virtually the entire lamellar region is non proliferative [30], and remodeling of the various cells of the secondary and primary lamellae occurs via controlled MMP activity [23]. The results of the present work seem to be aligned with those obtained by de Laat et al. [31], who showed MMP-2 to be secreted by basal cells and to be present in the lamellar tissue of healthy horses; however, the study by de Laat and colleagues did not show any increase in MMP gene expression, protein concentration, or activation in lamellar tissue taken from horses with insulin-induced laminitis, suggesting that it does not play a significant role in the development of this form of the disease. However, in the case of laminitis induced by starch overload or other causes, increases in MMP-2 have been detected and may be accompanied by the activation of other MMPS, such as MMP-9 [32].

MMP-14 may also play a key role in the maintenance of lamellar homeostasis under physiological conditions. One possibility is that MMP-14 and MMP-2 act simultaneously to remodel the non-proliferative basal cells of the inner hoof wall [30, 31]. MMP-14 has been shown to be regulated in breast cancer, where it is reported to activate MMP2 that is, in turn, associated with increased risk of malignancy and metastasis [33]. We cannot exclude the possibility that a similar mechanism occurs during laminitis, where MMP-14 activation may cause the subsequent activation of MMP-2 and thereby trigger the entire laminitic process.

The triggering factors able to induce laminitis have yet to be clearly identified. Although cellular and molecular events have been studied in the past, yielding important contributions to our current knowledge, much remains unknown about the early events of the disease process. One of the main known risk factors entails a metabolism-related pathology [4]. The accumulation of triosephosphate intermediates induced by an increase in reactive carbonyl species is caused by an enhanced metabolic flux [34]. Methylglyoxal belongs to a class of reactive carbonyl species known as the alpha-oxoaldehydes or dicarbonyls. It is derived from the spontaneous degradation of triosephosphate intermediates and acts as a highly potent glycating agent [34]. Dicarbonyls, such as methylglyoxal, can modify DNA and react with the amino groups of intracellular and extracellular proteins to form AGEs. Increased levels of AGEs induce

alterations in physiological cell cycles, leading to extreme stress conditions and even cell death. The interaction of AGE-modified proteins through cell surface receptors, such as RAGE, can lead to increased cellular activation and sustained inflammatory responses [35]. Significantly higher levels of the aforementioned factors have been clinically demonstrated in various pathologies; for example, erythrocytes from diabetic subjects show higher levels of triosephosphate intermediates and methylglyoxal compared with those of healthy subjects [6, 34].

In addition to the metabolic causes of laminitis, many other theories have been proposed, of which the production of bacterial exotoxins seems to be one of the most important. Indeed, a great deal of research has been conducted in the attempt to elucidate a possible bacterial role. However, a study by Reisinger et al. [19] demonstrated that lipopolysaccharide (LPS) extracted from bacteria *Escherichia coli* alone is not sufficient to induce laminitis *in vitro*. The majority of horses in the study by Tóth et al. [36] developed laminitis after the administration of a low dose of oligofructose in the presence of endotoxins: this strongly suggests that endotoxins are unlikely to induce laminitis on their own, more likely contributing to the disease's development and progression.

One drawback of *ex vivo* studies is that tissues are isolated from the whole organism, thus the complex processes occurring on the systemic level are missing. For example, in the case of hoof explants, the lack of cytokines might explain why in some experiments that used LPS, high concentrations were necessary to induce lamellar separation [25].

Moreover, the diffusion of nutrients and oxygen can vary throughout the explant and could depend on the explant dimensions (i.e., an excessive thickness could limit the permeability of the nutrients contained in the medium). To minimize all these effects, the present study employed a complex approach with several evaluations. In addition, in this study we have used male draft horses (Breton horses) that were relatively young; it would be interesting to apply the same *ex vivo* model for laminitis to even older horses of different breeds, such as pony breeds, to assess any differences in the studied parameters.

## Conclusions

The results of the present study provide evidence showing that it is possible to maintain HEs for at least 48 h following their harvest, and that sections could be used for *ex vivo* studies. The results of histological examination and the separation force test showed that increasing concentrations of MG lead to the deterioration of HE tissue integrity. Moreover, the effect of MG seems to be time -dependent because samples incubated for 48 h showed higher levels of deterioration compared with samples incubated for 24 h. RNA quantification of MMP-2 and -14 and TIMP-2 expression supported the occurrence of extracellular matrix remodeling induced by MG in all samples. Further studies are needed to advance our understanding of the role of glucose metabolism in pathogenesis of laminitis.

## Supporting information

**S1 Dataset. Datasets of the different experimental assays.**
(ZIP)

**S1 Fig. Step-by-step procedure. A:** Hooves obtained from a commercial abattoir were transported in ice to the lab, immediately cleaned using a hoof knife and brush and scrubbed. **B:** Hooves were trimmed using an electric table saw on the medial and lateral aspects to facilitate further cutting. Sagittal cuts were made to create 4–5 slices of the digit, measuring 0.8x1.5 cm in thickness containing 10–12 lamellae. **C:** Slices were trimmed using saw and scalpel blade to obtain 0.7cm X 0.5 cm blocks of tissue. Each block consists of inner hoof wall (a), epidermal

lamellae (b), dermal lamellae (c), and distal phalanx (d). Explants were rinsed and placed in a sterile saline solution. **D:** The lamellar explants were in Dulbecco's Modified Eagles Medium (DMEM) supplemented with % of antibiotic/antimycotic solution, 2% of L-glutamine, and 10% of Fetal Bovine Serum (FBS) (defined as DMEM+) at 37˚C and 5% $CO_2$.
(TIF)

## Acknowledgments

The preliminary study results were presented in the Equine Colic Research Symposium of 2014 (Dublin, Ireland) and in Sisvet Congress 2017 (Naples, Italy).

## Author Contributions

**Conceptualization:** Marco Gandini, Emanuela Valle.

**Data curation:** Cristina Vercelli, Massimiliano Tursi, Silvia Miretti, Gessica Giusto.

**Investigation:** Cristina Vercelli, Massimiliano Tursi, Silvia Miretti, Gessica Giusto, Marco Gandini.

**Methodology:** Emanuela Valle.

**Supervision:** Giovanni Re, Emanuela Valle.

**Writing – original draft:** Cristina Vercelli, Giovanni Re.

**Writing – review & editing:** Cristina Vercelli, Massimiliano Tursi, Silvia Miretti, Gessica Giusto, Marco Gandini, Giovanni Re, Emanuela Valle.

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
