## [Decision Letter · Decision Letter 0]

9 Apr 2021

PONE-D-20-06143

Effect of sugar metabolite methylglyoxal on horse keratinocytes: an ex vivo model for laminitis

PLOS ONE

Dear Dr. Vercelli,

Thank you for submitting your manuscript to PLOS ONE. After careful consideration, we feel that it has merit but does not fully meet PLOS ONE’s publication criteria as it currently stands. Therefore, we invite you to submit a revised version of the manuscript that addresses the points raised during the review process.

Reviewer 1 has identified a number of questions that need to be addressed regarding aspects of the methods and analyses included in your study. Please ensure that you provide detailed responses to each of these questions and concerns, as well as addressing the presentational issues raised.

We look forward to receiving your revised manuscript.

Kind regards,

Jamie Males

Senior Editor

PLOS ONE

Journal Requirements:

Thank you for stating in your Funding Statement:

The present study was funded with ex 60% fund of the University of Turin.

3. We noticed you have some minor occurrence of overlapping text with the following previous publications, which needs to be addressed:

- https://www.ncbi.nlm.nih.gov/pmc/articles/PMC3605361/

- https://onlinelibrary.wiley.com/doi/full/10.1002/dmrr.2811

- https://beva.onlinelibrary.wiley.com/doi/abs/10.2746/042516406778400565

- https://www.sciencedirect.com/science/article/abs/pii/S0165242711000201?via%3Dihub

The text that needs to be addressed involves:

- Lines 297-302

- Lines 303-310

- Lines 315-326

In your revision ensure you cite all your sources (including your own works), and quote or rephrase any duplicated text outside the methods section. Further consideration is dependent on these concerns being addressed.

Reviewers' comments:

Reviewer's Responses to Questions

**Comments to the Author**

1. Is the manuscript technically sound, and do the data support the conclusions?

Reviewer #1: Partly

Reviewer #2: Yes

2. Has the statistical analysis been performed appropriately and rigorously? 

Reviewer #1: Yes

Reviewer #2: Yes

3. Have the authors made all data underlying the findings in their manuscript fully available?

Reviewer #1: Yes

Reviewer #2: Yes

4. Is the manuscript presented in an intelligible fashion and written in standard English?

Reviewer #1: Yes

Reviewer #2: Yes

5. Review Comments to the Author

Reviewer #1: The authors have submitted a manuscript describing an ex vivo model of laminitis that they have created using a glucose metabolite (methylglyoxal) applied to lamellar explants. Such a model, if it could be validated, would be extremely useful for research in this area and has, in fact, been sought for some years now by researchers in the field. This could be a valuable contribution to those efforts - I have some questions and comments for the authors about their work, listed below (by line number in the manuscript):

Line 1 - 'Effect of the sugar metabolite...'; ex vivo should be italicized

Abstract:

Line 15 - '...synthesized during the digestion process in horses...'

Line 16 - '...excessive levels could lead to alterations in the hoof lamellar structure.'

Line 18 - '...were applied to hoof explants (HE), which...'

Line 19 - 'Microscopic'

Line 20 - '..post-MG application.'

Line 21 - '...(MMP)-2 and -14, and tissue inhibitor of metalloproteinases (TIMP)-2'

Line 22 - '...at each time point for all...'

Line 23 - '...mimicking laminitis. The separation force test revealed...'

Line 26 - '...significant weakness, and samples...'; 'In the same samples, high levels of MMP-2 and -14 and low levels of TIMP-2 were present.

Line 27 - 'All results support that high levels of MG could induce irreversible...'

Introduction:

Line 33 - 'It is characterized by damage/deterioration of the lamellar tissue...'

Line 37 - '...acute or chronic, and it is not usually readily reversible.'

Lines 38 - 39 - 'The most common etiology of laminitis cases involves gastrointestinal or metabolic disease.'

Line 41 - '...hormonal influences and weight/pressure are both involved..'

Line 46 - would add comma after 'isoforms'

Line 48 - '...disturbances resulting from ingestion...'

Line 49 - '...role in sepsis-related laminitis...'

Line 50 - '...used as highly digestible...'

Line 52 - '...glucose liberation...'

Line 54 - '...followed by colic and laminitis.' (founder is redundant here, I believe)

Line 56 - 'In this scenario, release of toxins from the hindgut is suspected to occur, which in theory induces degradation of the lamellar basement membrane...'

Line 58 - '...infiltration into the lamellae.'

Line 59 - '...can lead to rapid, devastating changes that result in the elaboration of toxins and other substances:...'

Line 60 - 'Gram-negative and -positive...'

Line 61 - '...stress, leading to ...'; has this substance been shown to appear in equine serum/plasma following induction of a carbohydrate overload model of laminitis? That would seem to be central to the premise here, given the proposed route of exposure of the lamellae to this substance in vivo.

Lines 64-65 - Would include some context-specific criteria where lactate may serve as a biomarker of laminitis risk (not laminitis itself) - certain situations are associated with hyperlactatemia but not laminitis risk (such as intense aerobic exercise), so would restrict this in some way to situations in which this is more likely to be the case (e.g., sepsis).

Line 67 - '...which precedes and enables remodeling...'

Line 70 - '...have shown that under specific...'

Line 72 - '...making them an attractive model for the...'

Line 77 - '...different exposure time points.'

Materials and Methods:

Line 81 - the authors might comment in the discussion about how widely these data might be extrapolated to other breeds/types of horses

Line 82 - '...were enrolled in the study.'

Line 84 - would add a comma after 'Italy)'; would remove the commas after 'collected' and 'consent'

Line 88 - would remove 'symptoms and' from this sentence; what signs of laminitis, specifically, were evaluated (would list them)? Also, would specify whether the limb was a front or hind limb.

Line 89 - would add a comma after 'joint'; '...within 45 minutes of slaughter and..'

Line 101 - would use 'lamellae' instead of 'laminae' consistently throughout

Line 114 - '48 hour incubation'

Lines 116-117 - would include how survival was evaluated here (not just in the Results section)

Line 117 - '...used for the rest of the study.'

Line 125 - '...integrity of the samples.'

Line 130 - '...lamellar site were considered valid measurements.'

Lines 130-131 - I'm not sure what the authors are trying to say here, it's a bit unclear; consider rewording this sentence.

Line 136 - '...then embedded in paraffin, and 3-um slices were mounted...'

Line 142 - 'epidermal'

Line 144 - '...or severe and extensive (Grade...'

Lines 146 - 147 - This sentence is unclear - what particular characteristic of the SEL was evaluated?

Line 149 - how long after the force testing did RNA extraction occur? Was RNA extracted from any samples that were not subjected to force testing? Can the authors comment on the likely influence of this testing on the mRNA concentrations of some of their target genes in the sample tissue (i.e., are they known to be influenced by stretch, if sufficient time had elapsed between force testing and RNA extraction)?

Line 157 - can remove the comma after 'USA)'

Line 162 - '...MMP-14, and TIMP-2 transcripts...'

Line 164 - 'Equus caballus'

Line 169 - '...gene accession numbers, and...'

Line 174 - '...(ACTB), and...'

Line 175 - how was this determined?

Line 177 - '...triplicate, and template...'

Line 179 - can remove comma after 'averaged'

Line 187 - 'All the data were analyzed with a commercially available software program...'

Lines 190-193 - 'The separation force test and q-PCR data were analyzed using one-way ANOVA and Tukey's...'

Line 192 - How were the distributions of the histology score data analyzed?

Line 196 - '...was accepted at values...'

Results:

Lines 200-201 - Were any more sensitive/quantifiable measures of autolysis used to evaluate this tissue? This seems like a very subjective assessment.

Line 202 - Can the authors include a figure displaying representative histologic sections?

Line 203 - '...in a few samples...'

Line 204 - 'lamellae'

Line 205 - '...changes, and histological...'

Line 206 - 'structures; all samples...'

Lines 208-209 - '...in the preliminary assay, samples were kept in DMEM+ for all subsequent experiments.'

Line 212 - '...on all samples...'

Line 213 - '...at predetermined time points.'

Line 215 - '...separation of the HE lamellar structures could be...

Line 221 - '...can be observed.'

Line 222 - 'However, these differences did not reach the level of statistical significance.'

Line 225 - 'methylglyoxal'; '(N=45 per time point)'

Line 235 - '...due to alteration of...'

Lines 237, 238, 241 - do the authors mean 'epidermal' instead of 'epithelial' in these instances?

Lines 251-252 - 'Nevertheless, these differences did not reach the level of statistical significance.'

Line 256 - would avoid commenting on trends, if possible

Line 266, 270 - 'MG' is missing in the last line of the legend for Figure 6 and Figure 7

Line 267 - 'Tissue inhibitor of metalloproteinases-2 (TIMP-2) gene...'

Discussion:

Line 274 - would add a comma after 'solution'; can the authors discuss any other papers that have attempted to characterize/use a lamellar explant model of laminitis, for the purposes of comparing their model to others? (I believe there are at least a few out there currently.)

Line 284 - '...but according to our results, FBS and L-glutamine are necessary for tissue preservation.'

Line 290 - '...hoof lamellae, rendering them less robust and unable...'

Line 291 - '...untreated samples are instead able to bear and remain intact.'

Line 301 - would add a comma after 'features'

Line 311 - would add a comma after 'concentration'

Line 315 - would remove comma after 'detected'

Line 320 - '...has been shown to be regulated in breast cancer, where...'

Line 321 - '...with increased risk of malignancy...'

Line 326 - '...in the past, yielding important...'

Line 333 - would remove comma after 'intermediates'

Line 341 - '...subjects show higher levels...'

Line 342 - '...compared with those of healthy subjects.'

Line 347 - 'Escherichia coli' - should be italicized

Line 348 - 'induce the laminitis process...'

Line 350 - '...suggests that endotoxins are unlikely to induce laminitis...'

Line 351 - 'more likely merely contribute to the...'

Lines 358-359 - can the authors clarify here how their approach and evaluations mitigated the effects of variable diffusion of nutrients and oxygen in explants?

Line 362 - '...following their harvest, and that sections could be used for ex vivo studies.'

Line 367 - would list specific MMP's here

Lines 367-368 - I don't think that MMP mRNA expression necessarily correlates well with enzymatic activity, so would probably refine this statement a bit.

Line 367-368 - 'Further studies are needed to advance our understanding...'

Figure 1 - 'Sample preparation'; 'Histology/histology'

Figure 2 - 'Outer'; 'Lamellae'

Figure 4 - 'Histological score'

Figures 5, 6, and 7 - 'Concentration of Methylglyoxal (mM)'

Reviewer #2: Congratulations for this important research work. It is not easy to culture hoof explants of horses mainly by secondary bacteria contamination. I only suggest you to replace the figure 1 by another one more explicative and less tedious. I also suggest to add a set of photographs showing the method for explant procurement in a sequential fashion (step by step). This technique is very important and useful for equine laminitis researchers around the world.

6. PLOS authors have the option to publish the peer review history of their article (what does this mean?). If published, this will include your full peer review and any attached files.

Reviewer #1: No

Reviewer #2: No

---

## [Author Response · Author response to Decision Letter 0]

3 May 2021

Response letter manuscript PONE-D-20-06143

• The manuscript has been formatted according to Journal’s style

• The specification about funding has been added and the previous funding statement has been corrected

• The overlapping texts have been corrected according to the suggestions. 

o https://www.ncbi.nlm.nih.gov/pmc/articles/PMC3605361/ � this reference (Miretti et al., 2017) has been added along the text and in reference list. Thank You for the suggestion because we forgot to cite a paper of one of the co-authors. 

o https://onlinelibrary.wiley.com/doi/full/10.1002/dmrr.2811 � this paper was already present in the reference list and we added a new reference along the text.

o https://beva.onlinelibrary.wiley.com/doi/abs/10.2746/042516406778400565 � we are really sorry but we can not add this reference because we did not consult it to write the paper. We don’t know why there’s an overlapping.

o https://www.sciencedirect.com/science/article/abs/pii/S0165242711000201?via%3Dihub � this paper was already present in the reference list and we added a new reference along the text.

Reviewer 1

Dear Reviewer, on behalf of all Authors, I would like to thank You for the time that you have spent to revise our manuscript. We appreciate Your efforts to improve our manuscript. We accepted all Your suggestions, and we provided the best answers that we could to the questions and comments that You addressed. 

Here I provide a point-to-point response. When rephrasing was suggested, I have just stated “fixed”. When a longer answer was needed, I provided an explanation. 

Line 1: fixed

Abstract

• Line 15: fixed

• Line 16: fixed

• Line 18: fixed

• Line 19: fixed

• Line 20: fixed

• Line 21: fixed

• Line 22: fixed

• Line 23: fixed

• Line 26: fixed 

• Line 27: fixed

Introduction

• Line 33: fixed

• Line 37: fixed

• Line 38-39: fixed

• Line 41: fixed

• Line 46: fixed

• Line 48: fixed

• Line 49: fixed

• Line 50: fixed

• Line 52: fixed

• Line 54: fixed

• Line 56: fixed

• Line 58: fixed

• Line 59: fixed

• Line 60: fixed

• Line 61: fixed. To answer to the Reviewer’s question “ has this substance been shown to appear in equine serum/plasma following induction of a carbohydrate overload model of laminitis? That would seem to be central to the premise here, given the proposed route of exposure of the lamellae to this substance in vivo” some sentences were added in order to clarify (L61-69 in the revised version). 

• Line 64-65: To answer to the Reviewer’s question “Would include some context-specific criteria where lactate may serve as a biomarker of laminitis risk (not laminitis itself) - certain situations are associated with hyperlactatemia but not laminitis risk (such as intense aerobic exercise), so would restrict this in some way to situations in which this is more likely to be the case (e.g., sepsis)” some sentences were added (L75-80 in the revised version).

• Line 67: fixed

• Line 70: fixed

• Line 72: fixed

• Line 77: fixed 

Material and methods

• Line 81: According to Reviewer’s comment: “the authors might comment in the discussion about how widely these data might be extrapolated to other breeds/types of horses”, Authors provided a short explanation in discussion session (L 389-391 in the revised version). 

• Line 82: fixed

• Line 84: fixed

• Line 88: To answer to the Reviewer’s questions: “hind limb or front limb? List signs of laminitis that were excluded?” Authors added that front limbs were collected (L 99 in the revised version) and wrote the evaluated signs and symptoms (L 103-104 in the revised version). 

• Line 89: fixed

• Line 101: fixed in this line and along the text

• Line 114: fixed

• Lines 116-117: According to Reviewer’s comment: “would include how survival was evaluated here (not just in the Results section)” a short sentence was added (L 131-132 in the revised version). 

• Line 117: fixed

• Line 130: fixed

• Lines 130-131: According to Reviewer’s comment: “I'm not sure what the authors are trying to say here, it's a bit unclear; consider rewording this sentence”, Authors rephrased (L 147-149 in the revised version). 

• Line 136: fixed

• Line142: fixed

• Line 144: fixed

• Lines 146-147: According to Reviewer’s comment: ”how long after the force testing did RNA extraction occur? Was RNA extracted from any samples that were not subjected to force testing? Can the authors comment on the likely influence of this testing on the mRNA concentrations of some of their target genes in the sample tissue (i.e., are they known to be influenced by stretch, if sufficient time had elapsed between force testing and RNA extraction)?” Authors added some short sentences in lines 147-150 and 169-171 in order to explain that samples were immediately treated after the separation force test. According with reviewer's comment could be interesting evaluate changing in gene expression after the force test, but when the samples are maintained in culture condition for additional time after the test as reported by other studies (REF: Wu JH et al, Bone Joint Res. 2017 Mar;6(3):179-185; Qin TW, Sun YL, Thoreson AR, et al.. Effect of mechanical stimulation on bone marrow stromal cell-seeded tendon slice constructs: a potential engineered tendon patch for rotator cuff repair. Biomaterials 2015;51:43-50.The revitalisation of flexor tendon allografts with bone marrow stromal cells and mechanical stimulation: An ex vivo model revitalising flexor tendon allografts). In our case, we consider negligible the influence of the force test on the gene expression due to the closed timing between the test and RNA extraction.

• Line 157: fixed

• Line162: fixed

• Line 164: fixed

• Line 169: fixed

• Line 174: fixed

• Line 175: According to Reviewer’s question “how was this determined?” Authors answer that GAPDH was selected as housekeeping gene for the stable Cq values among the different experimental conditions. The correction is provided in lines 194-197 of the revised version.

• Line 177: fixed

• Line 179: fixed

• Line 187: Authors tried to provide the suggested correction. We hope to have interpreted well the comments. 

• Line 190-193: fixed

• Line 192: According to Reviewer’s question: “How were the distributions of the histology score data analyzed?” the test’s name (L210-211 in the revised version) and a short explanation (L 269-270 in the revised version) were added. 

• Line 196: fixed

Results

• Lines 200-201: According to Reviewer’s comment: “Were any more sensitive/quantifiable measures of autolysis used to evaluate this tissue? This seems like a very subjective assessment.”, Authors tried to rephrase and explain better (L 221 in the revised version). 

• Line 202: According to Reviewer’s question “Can the authors include a figure displaying representative histologic sections?”, Authors provided pictures to show the differences between well preserved hoof explant samples (using DMEM+) and other samples that were considered autholitic and not eligible for the further experiments. Pictures are presented as Figure 3 (L 232-238 in the revised version). The other pictures have been renumbered as consequence. 

• Line 203: fixed

• Line 204: fixed

• Line 205: fixed

• Line 206: fixed

• Lines 208-209: fixed

• Line 212: fixed

• Line 213: fixed

• Line 215: fixed

• Line 221: fixed

• Line 222: Authors tried to provide the suggested correction. We hope to have correctly interpreted the comments.

• Line 225: fixed

• Lines 237, 238, 241: According to Reviewer’s question: “do the authors mean 'epidermal' instead of 'epithelial' in these instances?” the answer is: Yes. Authors provided a correction along the text (Lines 255, 256 and 259 in the revised version). 

• Lines 251-252: fixed

• Line 256: According to Reviewer’s comment: “would avoid commenting on trends, if possible”, only statistically significances were considered. 

• Line 266-270: fixed

• Line 267: fixed

Discussion

• Line 274: fixed. According to Reviewer’s comment: “can the authors discuss any other papers that have attempted to characterize/use a lamellar explant model of laminitis, for the purposes of comparing their model to others? (I believe there are at least a few out there currently.)”, short sentences were added (L325-329 in the revised version). 

• Line 284: fixed

• Line 290: fixed

• Line 291: fixed

• Line 301: fixed

• Line 311: fixed

• Line 315: fixed

• Line 320: fixed

• Line 321: fixed

• Line 326: fixed

• Line 333: fixed

• Line 341: fixed

• Line 342: fixed

• Line 347: fixed

• Line 348: fixed

• Line 350: fixed

• Line 351: fixed

• Line 358-359: According to Reviewer’s question “can the authors clarify here how their approach and evaluations mitigated the effects of variable diffusion of nutrients and oxygen in explants?” Authors added a short sentence to clarify (L380-381 in the revised version). 

• Line 362: fixed

• Line 367: fixed

• Lines 367-368: According to Reviewer’s comment: “I don't think that MMP mRNA expression necessarily correlates well with enzymatic activity, so would probably refine this statement a bit”, the sentence has been refined. 

• Lines 367-368: fixed

Reviewer 1 also requested the following modifications on Figures: 

• Figure 1 - 'Sample preparation'; 'Histology/histology’: Figure 1 has been changed and designed according to the Reviewer 2 comment, considering also the corrections proposed by Reviewer 1. 

• Figure 2 - 'Outer'; 'Lamellae’: Figure has been corrected according to the Reviewer’s comment

• Figure 4 - 'Histological score: Figure has been corrected according to the Reviewer’s comment

• Figures 5, 6, and 7 - 'Concentration of Methylglyoxal (mM)': all the Figures have been corrected according to Reviewer suggestion. 

Reviewer 2:

Reviewer 2 wrote: “Congratulations for this important research work. It is not easy to culture hoof explants of horses mainly by secondary bacteria contamination. I only suggest you to replace the figure 1 by another one more explicative and less tedious. I also suggest to add a set of photographs showing the method for explant procurement in a sequential fashion (step by step). This technique is very important and useful for equine laminitis researchers around the world”.

Dear Reviewer, we are glad that our paper seems interesting for You. On behalf of all Authors, I would like to thank You for the time that you have spent to revise our manuscript. We accepted all Your suggestions, and here I provide the answers to Your comments.

• According to Reviewer suggestion, Figure 1 was changes. Authors hope that this new design is easier to read and more attractive for readers. 

• Pictures: a panel of 4 pictures has been added as supplementary material ("Other") in order to show the different passages of the procedure.

---

## [Decision Letter · Decision Letter 1]

26 May 2021

PONE-D-20-06143R1

Effect of sugar metabolite methylglyoxal on horse keratinocytes: an ex vivo model for laminitis

PLOS ONE

Dear Dr. Vercelli,

Thank you for submitting your manuscript to PLOS ONE. After careful consideration, we feel that it has merit but does not fully meet PLOS ONE’s publication criteria as it currently stands. Therefore, we invite you to submit a revised version of the manuscript that addresses the points raised during the review process.

We look forward to receiving your revised manuscript.

Kind regards,

Kanhaiya Singh, Ph.D

Academic Editor

PLOS ONE

Journal Requirements:

Additional Editor Comments (if provided):

Please address the suggestions made by Reviewers 2 and 3.

Reviewers' comments:

Reviewer's Responses to Questions

**Comments to the Author**

1. If the authors have adequately addressed your comments raised in a previous round of review and you feel that this manuscript is now acceptable for publication, you may indicate that here to bypass the “Comments to the Author” section, enter your conflict of interest statement in the “Confidential to Editor” section, and submit your "Accept" recommendation.

Reviewer #1: (No Response)

Reviewer #2: All comments have been addressed

Reviewer #3: (No Response)

2. Is the manuscript technically sound, and do the data support the conclusions?

Reviewer #1: Yes

Reviewer #2: Yes

Reviewer #3: Yes

3. Has the statistical analysis been performed appropriately and rigorously? 

Reviewer #1: Yes

Reviewer #2: Yes

Reviewer #3: Yes

4. Have the authors made all data underlying the findings in their manuscript fully available?

Reviewer #1: Yes

Reviewer #2: Yes

Reviewer #3: Yes

5. Is the manuscript presented in an intelligible fashion and written in standard English?

Reviewer #1: Yes

Reviewer #2: Yes

Reviewer #3: Yes

6. Review Comments to the Author

Reviewer #1: The author have addressed the majority of the questions and comments that I had included in a review of the previous version of this manuscript, and I believe that it is improved. I have a few additional questions, by line, below:

Title - 'Effect of the sugar metabolite...'; would use 'equine lamellar explants' instead of 'horse keratinocytes' in the title, to more accurately describe the tissue that was used; '... model of laminitis'

Abstract:

Line 18 - '...and excessive levels absorbed into peripheral blood could be delivered to the foot and lead to...'

Line 23 - can remove the comma after '14'

Line 25 - can remove 'a' after 'mimicking'

Line 27-28 - this sentence (the one that begins with 'In the same...') is incomplete and should be reworded

Introduction:

Line 41 - '...resembling the human diabetic...'

Line 49 - please add a space between 'from' and 'ingestion'

Line 55 - '...caecal pH, often followed by colic and laminitis...'

Line 59 - '...AGEs accumulate in significant amounts in the hoof lamellar tissue in the acute...'

Line 61 - 'using the hyperinsulinemic model...'

Line 63 - '...MG) causes the formation of AGEs, leading...'

Line 66 - '...intermediate, it is converted...'

Line 72 - 'In equines, feeding large amounts of fermentable carbohydrates with subsequent acidosis increases the level of plasma D-lactate...'

Line 75 - '...D-lactate can be considered a...'

Line 84 - '...making them an attractive model...'

Line 87 - '...of MG on equine lamellar explants in an ex vivo...'

Materials and Methods:

Line 99 - '...typical stance, increased digital pulse, and increased hoof capsular warmth).'

Line 100 - '...the right fore distal limb...'

Line 112 - '...dermal lamellae, and the bone.'

Line 120 - can remove 'of' in front of 'Fetal'

Line 126 - 'Sample'

Line 142 - would use 'lamellae' instead of 'laminae' throughout the manuscript

Line 143 - 'broken at'

Line 149 - 'One sample from each time point and from each MG concentration...'

Line 150 - would add a comma after 'paraffin'; '...slices were mounted...'

Line 153 - '...microscope, and images were captured...'

Line 160 - '...evaluated for lesions of the secondary...'

Line 164 - 'RNA-Later solution (Ambion), and the stratum lamellatum was disrupted using...'

Line 177 - '...MMP-14, and TIMP-2 transcripts, qPCR...'

Line 179 - 'designed from Equus caballus...'

Line 200 - '...were designed from Equus caballus...'

Line 205 - 'one-way ANOVA and Tukey's multiple...

Line 207 - '...their distributions were analyzed using two-way ANOVA...'

Results and discussion:

Line 221 - '...all structures; all samples were...'

Line 225 - 'characteristics'

Line 236 - 'predetermined'

Line 245 - '...for 48 h can be observed.'

Line 258 - '...of all layers and were statistically...'

Line 267 - 'concentrations'

Line 280 - can remove the comma after 'concentration'

Line 296 - '...as a suitable viable model system for investigating...'

Line 307 - 'with our results, FBS and...'

Line 309 - '...period, supporting the viability of the samples...'

Line 310 - 'Other papers have evaluated the possibility of using hoof explant samples but focus their attention on different metabolic goals (e.g.,...'

Line 312 - 'To the authors' knowledge, this is the first paper focusing specifically on the role of MG (...'

Line 315 - 'to modification of the hoof lamellae...'

Line 316 - '...that untreated samples are able to bear...'

Line 317 - '...confirmed that higher concentrations of MG and a longer...'

Line 319 - '...were confirmed by histological...'

Line 322 - 'The results of gene expression of MMPs...'

Line 325 - '...matrix remodeling, more severe histological features, and...'

Line 337 - '...starch overload or other causes, increases...'

Line 346 - can remove the comma after 'MMP2' here

Line 349 - 'yielding'

Line 363 - '...various pathologies; for example,...'

Line 370 - '...not sufficient to induce laminitis in vitro.'

Line 373-374 - '...more likely contributing to the disease's...'

Line 383-385 - '...(Breton horses) that were relatively young; it would be interesting to apply the same ex vivo model of laminitis to even older horses of different breeds, such as pony breeds, to assess any differences...'

Line 388 - '...could be used for ex vivo studies.'

Line 393 - would use 'supported' instead of 'confirmed' here

Supplemental data:

Line 519 - would use 'lamellar' instead of 'laminar' here

Reviewer #2: Congratulations. This is an important research work. In general, you addressed the concerns raised during the review process.

Reviewer #3: In this paper, the authors are examining the effects of excessive methylglyoxal on the hoof lamellar structure with an aim to identify the mechanism behind the Laminitis disease. They use an ex vivo experimental design and perform various types of analysis - macroscopic analysis, histological analysis, separation force test and gene expression. They identify that high levels of methylglyoxal could induce irreversible damage in the hooves which mimicks laminitis in an ex vivo model.

Overall, the findings of the manuscript are well-supported by the data and the methods used are appropriate. The authors have also addressed the reviewers comments adequately. However, there are some points to consider that I have outlined below:

1. In the revised manuscript, the authors have renamed the ‘Results’ section as ‘Results and Discussion’, however, the discussion has not been incorporated with the results. It is still presented as a separate section (minus the heading). The layout of this section will have to be changed significantly if the authors want to keep the ‘Results and Discussion’ heading. If not, I would recommend adding the ‘Discussion’ heading and keeping ‘Results’ separate.

Minor concerns:

1. In the abstract, the authors mention ‘Microscopical and histological analysis’, however, in the rest of the manuscript, there is ‘Macroscopical analysis’. Please correct accordingly.

2. In the abstract, the second last line starting with ‘In the same samples…’ (Line 29), does not make sense, it seems incomplete. Please edit.

3. Line 45, ‘…influences and weight/ pressures…’ needs to be edited for coherence.

4. Lines 48-51, need to be changed to make sense. Right now, it seems like a collection of terms.

5. Line 57, ‘this means’ should be ‘which means’.

6. Line 64, ‘hoof lamellar tissue level’, the word ‘level’ can be removed.

7. Line 70, ‘highly reactive intermediate is’, should be ‘highly reactive intermediate, it is’.

8. Line 78, ‘sequent acidosis’, do the authors mean ‘subsequent acidosis’?

9. Line 79, ‘increase the level of plasma levels’, should be ‘increase the plasma levels of’.

10. Line 81, ‘also because’ is redundant. Use either also, or because.

11. Line 89, ‘attracting model’ should be ‘attractive model’.

12. Line 100, ‘for this study purpose’ can be rephrased to, ‘for the purposes of this study’.

13. Line 105, ‘typical pain position and digital pulse and hot hoof’, one ‘and’ can be removed.

14. Line 107, ‘within 45 minutes of slaughter’ should be ‘within 45 minutes’, since the sentence is starting with ‘following slaughter’.

15. Line 114, Mungall and Pollitt is missing an in-text citation.

16. Line 131, should be ‘Fig S1’ to make it consistent with the rest of the manuscript.

17. Line 133, do the authors mean ‘Sample survival’?

18. Line 141, for ‘controls (K)’, reference the figure number.

19. Line 144, ‘the’ before structural integrity should be removed.

20. Line 150, do the authors mean ‘broken’, instead of ‘brocken’?

21. Line 172, change RNA later to RNAlater (as per manufacturers label).

22. Line 198 and Line 202, move Cq definition from line 202 to line 198.

23. Line 207, table 1 is not in line with the text.

24. Line 211-214, do not make sense. Which softwares? Which companies? Please edit. Line 211, remove ‘a’ before ‘commercially’ if you are referring to more than 1 software.

25. Line 216, ‘Turkey’s’ should be ‘Tukey’s’.

26. Line 222, define SD.

27. Line 228, should be ‘performed for all samples’.

28. Line 232-233, separate the sentences. ‘All samples’ should be a different sentence.

29. Line 248, should be ‘predetermined’.

30. Line 254-255 is very confusing. Please clarify.

31. Line 271-272, ‘and resulted statistically different’, please clarify.

32. Line 277, what do you mean by definition?

33. Line 280, should be ‘concentrations’.

34. Fig 6-8 legend, the gene descriptions should be in the text and not in the figure legends.

35. Line 328-329, please rephrase for coherence.

36. Line 330, should be ‘In our knowledge’.

37. Line 340, please edit for coherence.

38. Line 344, edit spelling to ‘correlated’.

39. Line 345, should be ‘resulting in structural weakness’.

40. Line 367, please define the ‘entire process’ so as to make the sentence complete.

41. Line 369, edit spelling to ‘yielding’.

42. Line 378, the acronym AGEs does not need to be defined again.

43. Line 391, the word ‘bacteria’ should be before E.coli and not after.

44. Line 395, rephrase for coherence.

45. Line 403-406, rephrase for coherence.

46. Line 410, remove ‘an’ before ex vivo.

47. Line 412, time-dependent would be more appropriate than ‘time-related’.

48. Figure 1, The authors can consider changing ‘force test’ to ‘separation force test’. It would then be consistent throughout the manuscript.

49. Figure 4, table headings are missing.

50. Figure 4,6,7,8, the legend for time needs to be consistent throughout. Either 24h, 48h or T24h, T48h.

51. Figure 6-8, the Y-axis should be Cq instead of Ct, for consistency between figures and text.

7. PLOS authors have the option to publish the peer review history of their article (what does this mean?). If published, this will include your full peer review and any attached files.

Reviewer #1: No

Reviewer #2: No

Reviewer #3: No

---

## [Author Response · Author response to Decision Letter 1]

7 Jun 2021

Response to Reviewers

Journal requirements

Dear Editor, 

As required in your email, the following items are under submission: 

• Response to Reviewers

• Revised Manuscript with Track Changes

• Manuscript

• Cover letter 

An updated and more precise financial disclosure statement is presented in our cover letter. 

All Figures have been checked using PAGE online system and are submitted according to PAGE system revision. 

Tables have been formatted according to https://journals.plos.org/plosone/s/tables

References have been numbered along the text in order at first citation and reference list has been updated consequently. Two references have been delated from the reference list (Bierhaus et al., 2012 and Kyaw-Tanner et al., 2012) because they were not present in-text. The in-text citation of “Visser 2008” (L199) has been removed because it does not exist. References have been organized and formatted in Vancouver style using Zotero software. 

Reviewer comments

Reviewer 1

Authors would like to thank R1 for all the efforts made to improve our paper. Authors accepted all suggestions that have been addressed and corrections have been provided along the text. 

Reading once again the revised manuscript, we correct some minor issue that are listed at the end of this letter. 

Here we copied R1 comments (italics) and we provide the answers. 

Reviewer #1: The author have addressed the majority of the questions and comments that I had included in a review of the previous version of this manuscript, and I believe that it is improved. I have a few additional questions, by line, below:

Title - 'Effect of the sugar metabolite...'; would use 'equine lamellar explants' instead of 'horse keratinocytes' in the title, to more accurately describe the tissue that was used; '... model of laminitis': corrections have been provided according to the comments.

Abstract:

• Line 18 - '...and excessive levels absorbed into peripheral blood could be delivered to the foot and lead to...' � correction has been provided according to Reviewer’s comment.

• Line 23 - can remove the comma after '14'�correction has been provided according to Reviewer’s comment.

• Line 25 - can remove 'a' after 'mimicking' correction has been provided according to reviewer’s comment.

• Line 27-28 - this sentence (the one that begins with 'In the same...') is incomplete and should be reworded �according to reviewer’s comment the sentence has been erased. R3 suggested the same.

Introduction:

• Line 41 - '...resembling the human diabetic...'. � correction has been provided according to reviewer’s comment.

• Line 49 - please add a space between 'from' and 'ingestion' �Correction has been provided according to reviewer’s comment.

• Line 55 - '...caecal pH, often followed by colic and laminitis...'. � Correction has been provided according to reviewer’s comment.

• Line 59 - '...AGEs accumulate in significant amounts in the hoof lamellar tissue in the acute...'. � Correction has been provided according to reviewer’s comment.

• Line 61 - 'using the hyperinsulinemic model...' Correction has been provided according to reviewer’s comment.

• Line 63 - '...MG) causes the formation of AGEs, leading...' � Correction has been provided according to reviewer’s comment.

• Line 66 - '...intermediate, it is converted...': � Correction has been provided according to reviewer’s comment.

• Line 72 - 'In equines, feeding large amounts of fermentable carbohydrates with subsequent acidosis increases the level of plasma D-lactate...'. � Correction has been provided according to reviewer’s comment.

• Line 75 - '...D-lactate can be considered a...'. � Correction has been provided according to reviewer’s comment.

• Line 84 - '...making them an attractive model...’ �Correction has been provided according to reviewer’s comment.

• Line 87 - '...of MG on equine lamellar explants in an ex vivo...'. � Correction has been provided according to reviewer’s comment.

Materials and Methods:

• Line 99 - '...typical stance, increased digital pulse, and increased hoof capsular warmth).' � Correction has been provided according to reviewer’s comment.

• Line 100 - '...the right fore distal limb...’ � Correction has been provided according to reviewer’s comment.

• Line 112 - '...dermal lamellae, and the bone. � Correction has been provided according to reviewer’s comment.

• Line 120 - can remove 'of' in front of 'Fetal'. � Correction has been provided according to reviewer’s comment.

• Line 126 - 'Sample'. � Correction has been provided according to reviewer’s comment

• Line 142 - would use 'lamellae' instead of 'laminae' throughout the manuscript �Correction has been provided according to reviewer’s comment in this line and along the text and captions. 

• Line 143 - 'broken at'. � Correction has been provided according to reviewer’s comment. Also, R3 suggested the same correction.

• Line 149 - 'One sample from each time point and from each MG concentration...'. � Correction has been provided according to reviewer’s comment.

• Line 150 - would add a comma after 'paraffin'; '...slices were mounted...'. � Correction has been provided according to reviewer’s comment.

• Line 153 - '...microscope, and images were captured...'. � Correction has been provided according to reviewer’s comment.

• Line 160 - '...evaluated for lesions of the secondary...'. � Correction has been provided according to reviewer’s comment.

• Line 164 - 'RNA-Later solution (Ambion), and the stratum lamellatum was disrupted using...' � Correction has been provided according to reviewer’s comment. R3 suggested ‘RNAlater’ instead of 'RNA-Later’. We checked the manufactured label, and we wrote ‘RNAlater’. 

• Line 177 - '...MMP-14, and TIMP-2 transcripts, qPCR...'. � Correction has been provided according to reviewer’s comment.

• Line 179 - 'designed from Equus caballus...'. �Correction has been provided according to reviewer’s comment.

• Line 200 - '...were designed from Equus caballus...'. � Correction has been provided according to reviewer’s comment.

• Line 205 - 'one-way ANOVA and Tukey's multiple.... �. Correction has been provided according to reviewer’s comment.

• Line 207 - '...their distributions were analyzed using two-way ANOVA...'. � Correction has been provided according to reviewer’s comment.

Results and discussion:

• Line 221 - '...all structures; all samples were...'. � Correction has been provided according to reviewer’s comment.

• Line 225 - 'characteristics'. � Correction has been provided according to reviewer’s comment.

• Line 236 - 'predetermined'. � Correction has been provided according to reviewer’s comment.

• Line 245 - '...for 48 h can be observed. � Correction has been provided according to reviewer’s comment.

• Line 258 - '...of all layers and were statistically...' �Correction has been provided according to reviewer’s comment.

• Line 267 - 'concentrations'. � Correction has been provided according to reviewer’s comment.

• Line 280 - can remove the comma after 'concentration'. � Correction has been provided according to reviewer’s comment.

• Line 296 - '...as a suitable viable model system for investigating...’ � Correction has been provided according to reviewer’s comment.

• Line 307 - 'with our results, FBS and...' �Correction has been provided according to reviewer’s comment.

• Line 309 - '...period, supporting the viability of the samples...' � Correction has been provided according to reviewer’s comment.

• Line 310 - 'Other papers have evaluated the possibility of using hoof explant samples but focus their attention on different metabolic goals (e.g.,...'. � Correction has been provided according to reviewer’s comment.

• Line 312 - 'To the authors' knowledge, this is the first paper focusing specifically on the role of MG (...' � Correction has been provided according to reviewer’s comment.

• Line 315 - 'to modification of the hoof lamellae...’ � Correction has been provided according to reviewer’s comment.

• Line 316 - '...that untreated samples are able to bear...'. � Correction has been provided according to reviewer’s comment.

• Line 317 - '...confirmed that higher concentrations of MG and a longer...' � Correction has been provided according to reviewer’s comment.

• Line 319 - '...were confirmed by histological...'. �. Correction has been provided according to reviewer’s comment.

• Line 322 - 'The results of gene expression of MMPs...' � Correction has been provided according to reviewer’s comment.

• Line 325 - '...matrix remodeling, more severe histological features, and...' � Correction has been provided according to reviewer’s comment.

• Line 337 - '...starch overload or other causes, increases...' � Correction has been provided according to reviewer’s comment.

• Line 346 - can remove the comma after 'MMP2' here. Correction has been provided according to reviewer’s comment. 

• Line 349 - 'yielding' �Correction has been provided according to reviewer’s comment.

• Line 363 - '...various pathologies; for example,...' � Correction has been provided according to reviewer’s comment

• Line 370 - '...not sufficient to induce laminitis in vitro.' �. Correction has been provided according to reviewer’s comment.

• Line 373-374 - '...more likely contributing to the disease's...' � Correction has been provided according to reviewer’s comment.

• Line 383-385 - '...(Breton horses) that were relatively young; it would be interesting to apply the same ex vivo model of laminitis to even older horses of different breeds, such as pony breeds, to assess any differences...'. � Corrections have been provided according to reviewer’s comment.

• Line 388 - '...could be used for ex vivo studies.' � Correction has been provided according to reviewer’s comment.

• Line 393 - would use 'supported' instead of 'confirmed' here. � Correction has been provided according to reviewer’s comment.

Supplemental data:

• Line 519 - would use 'lamellar' instead of 'laminar' here. �The correction has been provided according to the comment in L519 and in Fig S1. 

Reviewer 2

Reviewer #2: Congratulations. This is an important research work. In general, you addressed the concerns raised during the review process.

Author would like to sincerely thank R2 for all the efforts made to improve our paper. We are pleased to hear that R2 is satisfied about the corrections that have been provided in the first revision round. 

Reading once again the revised manuscript, we correct some minor issue that are listed at the end of this letter. 

Reviewer 3

Authors would like to thank R3 for all the efforts made to improve our paper. Authors accepted all suggestions that have been addressed and corrections have been provided along the text. 

Reading once again the revised manuscript, we correct some minor issue that are listed at the end of this letter. 

Here we copied R3 comments (italics) and we provide the answers. 

Reviewer #3: In this paper, the authors are examining the effects of excessive methylglyoxal on the hoof lamellar structure with an aim to identify the mechanism behind the Laminitis disease. They use an ex vivo experimental design and perform various types of analysis - macroscopic analysis, histological analysis, separation force test and gene expression. They identify that high levels of methylglyoxal could induce irreversible damage in the hooves which mimicks laminitis in an ex vivo model.

Overall, the findings of the manuscript are well-supported by the data and the methods used are appropriate. The authors have also addressed the reviewers comments adequately. However, there are some points to consider that I have outlined below:

1. In the revised manuscript, the authors have renamed the ‘Results’ section as ‘Results and Discussion’, however, the discussion has not been incorporated with the results. It is still presented as a separate section (minus the heading). The layout of this section will have to be changed significantly if the authors want to keep the ‘Results and Discussion’ heading. If not, I would recommend adding the ‘Discussion’ heading and keeping ‘Results’ separate. � Thank you for your suggestions. Authors would like to maintain the two sections separated. For this reason, headings have been changed. 

Minor concerns:

1. In the abstract, the authors mention ‘Microscopical and histological analysis’, however, in the rest of the manuscript, there is ‘Macroscopical analysis’. Please correct accordingly � According to reviewer’s comment, correction has been provided. 

2. In the abstract, the second last line starting with ‘In the same samples…’ (Line 29), does not make sense, it seems incomplete. Please edit. � the sentence has been edited.

3. Line 45, ‘…influences and weight/ pressures…’ needs to be edited for coherence. � The sentence has been edited according to reviewer’s comment. 

4. Lines 48-51, need to be changed to make sense. Right now, it seems like a collection of terms. � sentence has been rephrased in order to be clearer 

5. Line 57, ‘this means’ should be ‘which means’. � Correction has been provided according to reviewer’s comment.

6. Line 64, ‘hoof lamellar tissue level’, the word ‘level’ can be removed. � ‘Level ‘has been removed, also according to R1 suggestion.

7. Line 70, ‘highly reactive intermediate is’, should be ‘highly reactive intermediate, it is’. � The same correction was proposed also by R1. Correction has been provided. 

8. Line 78, ‘sequent acidosis’, do the authors mean ‘subsequent acidosis’? � yes, thank you for your comment, Correction has been provided. 

9. Line 79, ‘increase the level of plasma levels’, should be ‘increase the plasma levels of’.�Correction has been provided according to reviewer’s comment.

10. Line 81, ‘also because’ is redundant. Use either also, or because. � Correction has been provided according to reviewer’s comment.

11. Line 89, ‘attracting model’ should be ‘attractive model’. �The same correction was proposed also by R1. Correction has been provided. 

12. Line 100, ‘for this study purpose’ can be rephrased to, ‘for the purposes of this study’. � The sentence has been rephrased according to the comment of R3. 

13. Line 105, ‘typical pain position and digital pulse and hot hoof’, one ‘and’ can be removed. � R1 asked to rephrase the sentence. We hope that R3 agrees with this version. 

14. Line 107, ‘within 45 minutes of slaughter’ should be ‘within 45 minutes’, since the sentence is starting with ‘following slaughter’. � thank you for your comment, Correction has been provided. 

15. Line 114, Mungall and Pollitt is missing an in-text citation. � All the in-text citations have been formatted according to the journal requirements and the reference list has been updated. According to this, also the comment about L114 has been fixed. 

16. Line 131, should be ‘Fig S1’ to make it consistent with the rest of the manuscript. � correction has been provided. Thank you for the suggestion.

17. Line 133, do the authors mean ‘Sample survival’? � The same correction was proposed also by R1. Correction has been provided. 

18. Line 141, for ‘controls (K)’, reference the figure number. � reference to figures has been added

19. Line 144, ‘the’ before structural integrity should be removed. � ‘the’ has been removed 

20. Line 150, do the authors mean ‘broken’, instead of ‘brocken’? yes, thank you for your suggestion.

21. Line 172, change RNA later to RNAlater (as per manufacturers label). � R1 proposed a different correction (‘RNA-Later’) but we checked the manufacturer label and we accepted your suggestion. 

22. Line 198 and Line 202, move Cq definition from line 202 to line 198. � The Cq definition has been moved and the sentences have been rephrased in order to be clearer. 

23. Line 207, table 1 is not in line with the text. � all tables have been formatted according to the journal style. We hope that table 1 now is in line with the text.

24. Line 211-214, do not make sense. Which softwares? Which companies? Please edit. Line 211, remove ‘a’ before ‘commercially’ if you are referring to more than 1 software. � thank you for your suggestion. The sentence has been edited and corrections have been provided. 

25. Line 216, ‘Turkey’s’ should be ‘Tukey’s’. � Correction has been provided.

26. Line 222, define SD. � SD has been defined.

27. Line 228, should be ‘performed for all samples’. � thank you for your suggestion. Correction has been provided. 

28. Line 232-233, separate the sentences. ‘All samples’ should be a different sentence. � R1 asked to change ‘,’ with ‘;’ to separate the sentences. We hope that R3 agrees with R1.

29. Line 248, should be ‘predetermined’. � The same correction was proposed also by R1. Correction has been provided. 

30. Line 254-255 is very confusing. Please clarify. � these two lines have been modified according to the comments provided by reviewers in the fist round. Nevertheless, Authors checked the first manuscript that was submitted and tried to rephrase the sentences in order to be clearer and more related to the original meaning. 

31. Line 271-272, ‘and resulted statistically different’, please clarify. � sentence has been rephrased according to R1 comment. We hope that R3 agrees with R1.

32. Line 277, what do you mean by definition? � no, in this case, we mean the definition of the histological image.

33. Line 280, should be ‘concentrations’. � ‘s’ has been added according to R1 and R3comments. 

34. Fig 6-8 legend, the gene descriptions should be in the text and not in the figure legends. � the definitions have been removed from figure captions.

35. Line 328-329, please rephrase for coherence. � some corrections have been provided according to the R1 comments. We hope that R3 agrees with R1.

36. Line 330, should be ‘In our knowledge’. ��some corrections have been provided according to the R1 comments. We hope that R3 agrees with R1.

37. Line 340, please edit for coherence. � some corrections have been provided according to the R1 comments. We hope that R3 agrees with R1.

38. Line 344, edit spelling to ‘correlated’. � thank you. Correction has been provided. 

39. Line 345, should be ‘resulting in structural weakness’. �thank you for your comment. Correction has been provided. 

40. Line 367, please define the ‘entire process’ so as to make the sentence complete. � we add ‘laminitic’

41. Line 369, edit spelling to ‘yielding’. � also R1 suggested this correction. Correction has been provided. 

42. Line 378, the acronym AGEs does not need to be defined again. � definition has been removed.

43. Line 391, the word ‘bacteria’ should be before E.coli and not after. Thank you for your comment. This correction has been fixed.

44. Line 395, rephrase for coherence. � also R1 suggested a rephrasing here. We hope that R3 agrees with R1. 

45. Line 403-406, rephrase for coherence. � also R1 suggested a rephrasing here. We hope that R3 agrees with R1. 

46. Line 410, remove ‘an’ before ex vivo. � ‘an’ as been delated according to R1 and R3 suggestions.

47. Line 412, time-dependent would be more appropriate than ‘time-related’. � the word has been changed according to R3 suggestion.

48. Figure 1, The authors can consider changing ‘force test’ to ‘separation force test’. It would then be consistent throughout the manuscript. � Figure 1 has been changed according to R3 suggestion.

49. Figure 4, table headings are missing. � the complete table has been inserted. 

50. Figure 4,6,7,8, the legend for time needs to be consistent throughout. Either 24h, 48h or T24h, T48h. � Fig. 4,6-8 are now consistent and the same wording (T24 h and T 48 h) has been used along the text.

51. Figure 6-8, the Y-axis should be Cq instead of Ct, for consistency between figures and text. � Y axis label has been corrected according to R3 comment.

Authors comments

• Reading once again the manuscript, we have found some typos and double-spaced words: we have fixed everything. 

• Some references have been delated. An explanation is provided at the beginning of this letter and in the cover letter. 

• L292: Fig 9 does not exist. Thus a correction has been provided.

• L305: ‘/maintaing’ has been removed.

• L356: ‘ alpha oxoaldehydes’ instead of ‘alphaoxoaldehydes’ 

• In the whole document, MMP-2, MMP-14 and TIMP-2 (using ‘-‘) instead of MMP2, MMP14 and TIMP2 have been written. This was done in order to be coherent along the text, tables and figures using the same wording and according to a suggestion of the first round of revision. 

• Table 2: the title was missing. We added it.

---

## [Editor Report · Decision Letter 2]

15 Jun 2021

Effect of sugar metabolite methylglyoxal on equine lamellar explants: an ex vivo model of laminitis

PONE-D-20-06143R2

Dear Dr. Vercelli,

We’re pleased to inform you that your manuscript has been judged scientifically suitable for publication and will be formally accepted for publication once it meets all outstanding technical requirements.

Kind regards,

Kanhaiya Singh, Ph.D

Academic Editor

PLOS ONE
---

## [Editor Report · Acceptance letter]

19 Jul 2021

PONE-D-20-06143R2 

Effect of sugar metabolite methylglyoxal on equine lamellar explants: an ex vivo model of laminitis 

Dear Dr. Vercelli:

I'm pleased to inform you that your manuscript has been deemed suitable for publication in PLOS ONE. Congratulations! Your manuscript is now with our production department. 

Kind regards, 

on behalf of

Dr. Kanhaiya Singh 

Academic Editor

PLOS ONE